# Evaluation of Cognitive Function in Trials Testing New-Generation Hormonal Therapy in Patients with Prostate Cancer: A Systematic Review

**DOI:** 10.3390/cancers12092568

**Published:** 2020-09-09

**Authors:** Laura Marandino, Francesca Vignani, Consuelo Buttigliero, Teresa Gamba, Andrea Necchi, Marcello Tucci, Massimo Di Maio

**Affiliations:** 1Department of Oncology, University of Turin, 10043 Torino, Italy; laura.marandino@istitutotumori.mi.it (L.M.); consuelo.buttigliero@unito.it (C.B.); teresa.gamba@edu.unito.it (T.G.); 2Medical Oncology, Istituto Nazionale dei Tumori, Fondazione IRCCS, 20133 Milano, Italy; andrea.necchi@istitutotumori.mi.it; 3Medical Oncology, Ordine Mauriziano Hospital, 10128 Torino, Italy; fvignani@mauriziano.it; 4Medical Oncology, San Luigi Gonzaga Hospital, 10043 Orbassano, Italy; 5Medical Oncology, Cardinal Massaia Hospital, 14100 Asti, Italy; mtucci@asl.at.it

**Keywords:** cognitive function, prostate cancer, abiraterone, enzalutamide, apalutamide, darolutamide

## Abstract

**Simple Summary:**

In patients with prostate cancer, the use of new-generation hormonal therapy, added to androgen deprivation therapy, requires careful evaluation of cognitive function. The aim of this systematic review is to describe the evidence about cognitive function in randomized trials testing new-generation hormonal therapy (abiraterone, enzalutamide, apalutamide, darolutamide). For each trial, we assessed the availability of both investigator-assessed cognitive impairment and disorders and patient-reported evaluation of cognitive function. Out of 19 trials, the investigator-based evaluation of cognitive impairment was available in seven (36.8%), while patient-reported evaluation of cognitive function results was presented only in one trial (5.3%). This analysis shows that, despite cognitive deterioration could be relevant in patients with prostate cancer, clinical development of new-generation hormonal drugs has not included a systematic evaluation of cognitive function.

**Abstract:**

In patients with prostate cancer, earlier use and longer duration of new-generation hormonal therapy (NGHT), added to androgen deprivation therapy, requires careful evaluation of cognitive function. The aim of this systematic review is to describe the evidence about cognitive function in all the randomized trials (RCTs) testing NGHT (abiraterone, enzalutamide, apalutamide, darolutamide). We assessed the availability of both investigator-assessed cognitive impairment and disorders and patient-reported evaluation of cognitive function. Nineteen RCTs (17,617 patients) were included. The investigator-based evaluation of cognitive impairment was available in seven RCTs (36.8%). In total, 19/19 RCTs (100%) included patient-reported outcomes (PROs) collection, but PRO tools adopted allowed evaluation of cognitive function in two RCTs (10.5%). Among them, PRO-based cognitive function results were presented only in one RCT (5.3%): in ENZAMET, mean changes from baseline were worse with enzalutamide than with placebo, but deterioration-free survival favored enzalutamide. Despite cognitive deterioration could be relevant, clinical development of NGHT has not included a systematic evaluation of cognitive function. Assessment by investigators is at risk of underreporting, and commonly used PROs do not allow proper cognitive function analysis. Furthermore, the methodology of analysis can jeopardize the interpretation of results. Although direct comparisons are scanty, there could be differences between different NGHTs.

## 1. Introduction

Patients with cancer may experience cognitive problems as a consequence of their treatment [1]. Androgen deprivation therapy (ADT) is the mainstay of systemic treatment for patients with prostate cancer. Patients are usually treated for several years with ADT, which will be continued even during the castration-resistant phase of their disease. In healthy older subjects, lower testosterone levels have been found to be associated with worse cognitive functioning [2]. Cognitive impairment could have relevant clinical implications, because, in addition to memory and cognitive function alterations per se, it could imply loss of independence, increased incidence of falls with the associated risk of fractures and greater need for medical services [3]. Overall, the results of studies trying to describe the association between ADT and both cognitive changes and other CNS effects in patients with prostate cancer are controversial [3]. However, although the evidence about cognitive impairment in patients receiving ADT is variable, several studies have shown that ADT can produce a negative effect on cognitive functioning [4,5,6]. Patients receiving ADT could experience significant impairment on visuomotor tasks, while the differences found in the other cognitive domains (attention/working memory, executive function, language, verbal memory, visual memory, visuospatial ability) are less clear [4]. Impairment can already be evident 6–12 months after the beginning of treatment [6].

In addition to ADT, taxane-based chemotherapy, which is commonly used in patients with metastatic castration-resistant prostate cancer (CRPC) and, more recently, also in the hormone-sensitive setting, could have a negative impact on cognitive function [7].

Several new-generation hormonal therapies (NGHTs: abiraterone acetate, enzalutamide, apalutamide and darolutamide) have demonstrated efficacy, within randomized controlled trials, for patients with prostate cancer. Initially, these agents have been tested as a treatment for metastatic castration-resistant prostate cancer (CRPC), and subsequently for nonmetastatic CRPC and even for hormone-sensitive diseases. This anticipation of clinical use implies a progressively longer administration. For instance, median progression-free survival (PFS), grossly corresponding to the duration of treatment, for abiraterone acetate + prednisone was 5.6 months when used in the postchemotherapy CRPC setting [8], 16.5 months when used in the chemotherapy-naive CRPC setting [9], and 33.0 months when used in castration-sensitive patients [10]. With such a long hormonal treatment, added up to ADT, a good knowledge of the impact on patients’ quality of life (QoL), and particularly on cognitive function, has clinically relevant implications.

The aim of this review is to summarize the available patient-reported outcomes (PROs) results, in terms of cognitive function, in all randomized trials testing NGHT. We will focus on the importance of the evaluation of cognitive function in this setting, reviewing the available validated instruments, and discussing the adequacy of each instrument. In addition, we will discuss the opportunities and limitations of the different modalities of cognitive functioning data analysis and presentation.

## 2. Investigator-Assessed Cognitive Impairment

Treatment-related adverse events are commonly assessed, within clinical trials, by Common Terminology Criteria for Adverse Events (CTCAE) [11]. The current version (version 5.0) was released in November 2017, while previous versions (versions 3.0 and 4.0) were released in August 2006 and June 2010, respectively. All these versions include, within the category of nervous system disorders, cognitive disturbance (MedDRA Code 10009845), concentration impairment (MedDRA Code 10010250) and memory impairment (MedDRA Code 1002715). For all these adverse events, Grade 1 is mild, Grade 2 is moderate and Grade 3 is severe. Version 3.0 also included Grade 4 in the case of life-threatening severe clinical conditions due to cognitive impairment.

Despite the application of CTCAE could allow an accurate description of the incidence of cognitive function impairment, description by investigators of adverse events experienced by patients is clearly exposed to the risk of underreporting [12]. This risk could be particularly high for cognitive impairment, especially if mild or moderate, if not systematically assessed with adequate instruments during the clinical visits. Inclusion of QoL among study endpoints and adoption of PROs would be useful to reduce the underreporting of cognitive impairment.

## 3. PROs Commonly Used in Clinical Trials of Prostate Cancer

QoL evaluation is not performed, or not published, in a not negligible proportion of trials performed in prostate cancer [13]. In a systematic review of 72 randomized Phase 3 trials testing anticancer drugs in prostate cancer, issued between 2012 and 2018, the most common tools adopted for QoL evaluation were Functional Assessment of Cancer Therapy-Prostate (FACT-P), European Organisation for Research and Treatment of Cancer Quality of Life Questionnaire-Core 30 (EORTC QLQ-C30) and EORTC-Prostate 25 item (EORTC QLQ-PR25). Namely, FACT-P was adopted in the majority of trials conducted in the CRPC setting, while EORTC tools were employed in the majority of trials conducted in early stages or metastatic hormone-sensitive patients.

EORTC QLQ-C30 is among the PRO measures (PROM) commonly used to assess HRQoL in patients with cancer [14]. It contains five functional scales, including cognitive functioning. Specifically, it includes two questions about cognitive functioning: question 20 (“Have you had difficulty in concentrating on things, like reading a newspaper or watching television?”) and question 25 (“Have you had difficulty remembering things?”). Each question has four possible answers (not at all, a little, quite a bit, very much).

EORTC QLQ-PR25 is specifically designed to explore symptoms related to prostate cancer and its treatment [15]. However, it does not include questions about cognitive function.

FACT-P consists of the FACT-General (FACT-G) plus a prostate-specific subscale [16]. Namely, FACT-G is a reliable, valid, commonly used questionnaire for assessing QoL in patients with cancer, used for more than 20 years [17]. It consists of 27 items, divided into four subscales: physical, social, emotional and functional well-being. The prostate cancer subscale (PCS) is made of 12 items, related to prostate-specific questions, including sexuality, bowel/bladder function and pain. Unfortunately, neither FACT-G nor the PCS included in FACT-P directly assess cognitive functioning.

In conclusion, in most trials evaluating systemic treatments in patients with prostate cancer, cognitive function is not assessed at all, or it is assessed only by the means of the two questions included in the EORTC QLQ-C30.

## 4. Available Validated Instruments for Evaluation of Cognitive Function

The Montreal Cognitive Assessment (MoCA) is a rapid screening tool for cognitive impairment [18]. Although it has been designed to help in the detection of mild cognitive impairment and Alzheimer’s disease, it has been widely used to quickly assess cognitive status in many different clinical settings, including prostate cancer [19]. It cannot be considered a PROM. In fact, it consists of a 30-point test administered in 10 minutes to patients, who are asked to perform some tasks aiming at the evaluation of short-term memory, visuospatial abilities, executive functions, working memory, language and orientation to time and place [18].

Many other tools for the evaluation of multiple aspects of cognitive function by investigators exist, such as the Grober–Buschke test, Digit span forward and backward (WAIS-III), Code (WAIS-III), Trail-Making test, Doors test, Stroop Victoria, Verbal fluencies, Rey–Osterrieth Complex Figure, Number location (VOSP), Years of education and fNART [20]. Interestingly, an ongoing study, the Cog-PRO trial (NCT02907372), will assess these tools, in addition to the MoCA test, to evaluate the impact of new-generation hormonal treatments on the cognitive function of patients ≥70 years old with advanced CRPC.

The Functional Assessment of Cancer Therapy-Cognitive (FACT-Cog) questionnaire is a PROM designed to assess the perceived cognitive function and the impact on QoL in patients with cancer [21,22,23]. FACT-Cog is a 37-item member of the FACIT suite of questionnaires (Functional Assessment of Chronic Illness Therapy). The FACT-Cog includes four subscales: perceived cognitive impairments (18 items); perceived cognitive abilities (seven items); impact of perceived cognitive impairment on QoL (four items); and comments from others on cognitive function (four items). The FACT-Cog has been found to be reliable and valid, and it has been used in various cancer populations [21]. According to the FACIT’s recommended scoring method, the four subscales can be scored separately.

## 5. Different Modalities of Data Analysis and Presentation of PROs: Opportunities and Limitations

As for other PROs items, different, nonmutually exclusive methods can be adopted for the complete analysis and presentation of cognitive function results.

All analyses of cognitive function should be explicit, with appropriate references, the definition of minimum clinically important difference (MCID) for all the cognitive function scales analyzed. The definition of MCID is relevant for all the modalities of analysis and interpretation: MCID allows the discussion of the clinical relevance of mean changes from baseline and mean differences between arms. Furthermore, identification of MCID is essential to define the threshold for dividing patients into responders/nonresponders (or worseners/nonworseners when the aim is to describe a potential deterioration, as in the case of cognitive function), and it is also essential to define the occurrence of the event (i.e., a clinically meaningful worsening) for time-to-deterioration analysis.

Description of mean scores or mean differences from baseline at each time point allows a comparison of the trajectory of cognitive function among groups. However, this method does not necessarily capture a potentially relevant heterogeneity in the change experienced by individual patients [13,24]. For instance, if a minority of subjects will experience a relevant deterioration, while the majority will experience no change, the overall mean change could be judged as trivial and negligible (remaining under the threshold of clinical relevance), therefore leading to underestimation of the clinical impact of the cognitive worsening experienced by the minority. From this point of view, the so-called analysis of responders/worseners helps to define the proportion of subjects with clinically relevant effects within each group.

Let us hypothesize two treatment groups (each of 100 patients), with Drug A not producing any change in cognitive function and Drug B producing a relevant worsening (e.g., 20 points on a 0–100 scale) in a “sensitive” subgroup (10% of the patients), without any change in the remaining 90%. Analysis of the mean changes will describe a negligible mean change with Drug B (two points, which are the mean of 20 points in 10 subjects + 0 points in 90 subjects), and mean change 0 points with Drug A. Although statistically significant (Mann–Whitney test *p* = 0.001), if the MCID for these items is 5, this difference will be discussed as negligible and not clinically relevant. However, if we consider the proportion of worseners, it will be 10% with Drug B vs. 0% with Drug A (Chi-square *p* = 0.001), and this information will better describe the clinical impact in the two groups.

Although cognitive impairment could occur relatively early after the start of the treatment, a complete description of the trajectory of cognitive function for the entire duration of treatment is needed. Indeed, this concept particularly applies to prostate cancer, where new-generation hormonal agents have been recently tested in earlier settings compared to their initial demonstration of efficacy. Consequently, treatment duration and life expectancy of patients can be particularly long, and a proper evaluation of the benefits and harms associated with treatments should also consider long-term effects. Curves showing time-to-deterioration of cognitive function can be useful to obtain a description of any clinically relevant worsening, either early or late. In fact, time-to-deterioration is an important modality of longitudinal analysis of PROs that, as well as for other functional scales, symptoms and global QoL, can be useful to describe the trajectory of cognitive function [25]. To be performed correctly, time-to-deterioration analysis should treat other events (including deaths or clinical progressions) as competing events, and not as events considered as cognitive worsening.

As a matter of fact, no single method can be considered optimal and exhaustive for the description of the cognitive function trajectory. Ideally, more methods could be planned in the study protocol and presented in the publication. However, the space dedicated to QoL is often marginal, and secondary QoL papers, when existing, are often characterized by a long delay in the publication [26].

## 6. Methods: Literature Search

The search was performed in June 2020, to identify all randomized trials testing NGHT (abiraterone acetate, enzalutamide, apalutamide, darolutamide) in patients with prostate cancer. The literature search was performed using PubMed. The following keywords were used for a first search: *abiraterone AND random*; enzalutamide AND random*; apalutamide AND random*; darolutamide AND random**. In addition, in order to include in the review also the evidence about cognitive function produced within nonrandomized studies with the same drugs, we added a second PubMed search: *abiraterone AND cognitive; enzalutamide AND cognitive; apalutamide AND cognitive; darolutamide AND cognitive*. References of the selected articles were also checked to identify further eligible trials. Furthermore, proceedings of the main International meetings (American Society of Clinical Oncology (ASCO) annual meeting, European Society of Medical Oncology (ESMO) annual meeting), were searched from 2010 onwards for updated results and/or presentation of endpoints missing in the publications of all the identified trials.

We excluded trials testing NGHT in settings different from approved indications. Trials with NGHT as control arm were excluded, with the exception of the CARD trial, designed with the aim of comparing two treatment strategies (cabazitaxel vs. abiraterone/enzalutamide), both already approved for use in clinical practice in that setting [27]. Trials testing experimental combinations and trials evaluating the efficacy of other experimental treatments were excluded, considering that the evaluation of cognitive function in patients receiving NGHT was reasonably beyond trial objectives.

## 7. Trials Testing Abiraterone, Enzalutamide, Apalutamide, Darolutamide: Available Data on Cognitive Function

Overall, 19 RCTs (17,617 patients, 9988 assigned to new-generation hormonal treatments) were included in the analysis. As shown in Figure 1A, investigator-based evaluation of the incidence of cognitive impairment was available in seven RCTs (36.8%): one trial in the setting of mCRPC, three in nmCRPC and three in HSPC. In total, 19/19 (100%) included PROs collection, but the adopted PRO tools allowed evaluation of cognitive function only in two RCTs (10.5%). Among them, as shown in Figure 1B, PRO-based cognitive function results were presented only in one RCT (5.3%). The details of trials analyzed for each drug are reported below.

## 8. Trials Testing Abiraterone

Abiraterone acetate has an anti-androgenic mechanism of action. It is converted in vivo to abiraterone, which inhibits 17 a-hydroxylase/C17,20-lyase (CYP17), an enzyme required for androgen biosynthesis.

As shown in Table 1 and Table 2, eight RCTs testing abiraterone acetate (6383 patients, 3359 assigned to abiraterone) were eligible for the analysis: COU-AA-301 [8,28], COU-AA-302 [9,29], NCT01695135 [30], NCT01591122 [31], STAMPEDE [32], LATITUDE [10,33], CARD [27] and NCT02125357 [34]. Investigator-based evaluation of the incidence of cognitive impairment was available in two RCTs (25.0%): one trial in the setting of mCRPC and one in HSPC. All RCTs (100%) included PROs collection. However, the adopted PRO tools allowed evaluation of cognitive function in one RCT (12.5%) and, to date, PRO-based cognitive function results have not been presented yet. The details of each trial included in the analysis are described below, with the exception of two trials testing both abiraterone and enzalutamide [27,34], which are described in the next sections.

In the double-blind, randomized, Phase 3 COU-AA-301 trial, patients with metastatic CRPC progressed after docetaxel were randomly assigned to receive abiraterone acetate plus prednisone or placebo plus prednisone in addition to ADT [8,28]. After a median follow-up (mFU) of 20 months, 16% of patients in the abiraterone group were still on study. Abiraterone was associated with a significant improvement in overall survival (OS).

In the double-blind, randomized, Phase 3 COU-AA-302 trial, asymptomatic or mildly symptomatic patients with metastatic chemotherapy-naive CRPC were randomly assigned to receive abiraterone acetate plus prednisone or placebo plus prednisone in addition to ADT [9,29]. After a mFU of 49 months, treatment with abiraterone acetate was ongoing for 8% of patients. Abiraterone acetate plus prednisone showed a significant benefit in radiographic progression-free survival (rPFS) and OS.

In both the COU-AA-301 and COU-AA-302 trials, there are no details about the incidence of investigator-assessed cognitive disorders. PROMs adopted in both studies do not allow evaluation of cognitive function [35,36].

A double-blind, randomized, placebo-controlled Phase 3 study (NCT01695135) tested abiraterone acetate + prednisone in addition to ADT in Chinese patients with metastatic CRPC, after docetaxel failure [30]. Abiraterone acetate plus prednisone showed a significant improvement in time to PSA progression. In this study, there are no details about investigator-assessed cognitive disorders, and PROMs adopted do not allow evaluation of cognitive function. Another double-blind, randomized, placebo-controlled Phase 3 study (NCT01591122) tested abiraterone acetate + prednisone in addition to ADT in chemotherapy-naïve, asymptomatic or mildly symptomatic patients from China, Malaysia, Thailand and Russia with metastatic CRPC [31]. After a mFU of four months, 88% of patients were still receiving study treatment in the abiraterone group. Abiraterone acetate plus prednisone showed a significant improvement in time to PSA progression over the placebo plus prednisone group. In the NCT01591122 trial, there are no details about the incidence of investigator-assessed cognitive disorders. PROMs adopted in the study do not include an evaluation of cognitive function.

In the open-label, randomized, STAMPEDE trial, patients with locally advanced or metastatic HSPC were assigned to receive ADT alone or ADT plus abiraterone acetate and prednisolone [32]. At a mFU of 40 months, 28% of patients in the abiraterone group were still in treatment. Abiraterone acetate and prednisolone in addition to ADT showed significantly better OS compared to ADT alone. Investigator-assessed cognitive disturbance of any grade (assessed with the use of the CTCAE, initially version 3.0 and later, version 4.0) was reported in 6.4% of patients in the abiraterone arm versus 3.8% in the control arm, and Grade 3–4 cognitive disturbance in 0.4% of patients in the experimental arm versus 0.2% in the control arm [32]. QoL was evaluated with EORTC C30 and EORTC P25 questionnaires. However, even if EORTC C30 includes evaluation of cognitive functioning, the results about this functional scale have not yet been reported [37].

In the double-blind, randomized, Phase 3 LATITUDE trial patients with newly diagnosed high-risk metastatic HSPC were assigned to receive abiraterone acetate plus prednisone and ADT versus placebos plus ADT [10,33]. After a mFU of 51.8 months, treatment in the abiraterone acetate plus prednisone group was ongoing in 26% of patients [33]. The combination of abiraterone acetate plus prednisone with ADT showed significantly longer OS. The incidence of investigator-assessed cognitive disorders is not described in this trial, and the PROMs adopted in the study do not include an evaluation of cognitive function [38].

## 9. Trials Testing Enzalutamide

Enzalutamide is a competitive androgen-receptor inhibitor that blocks androgen-receptor nuclear translocation, recruitment of androgen-receptor cofactors and androgen-receptor binding to DNA [42].

As shown in Table 2 and Table 3, 10 RCTs testing enzalutamide (7923 patients, 4343 assigned to enzalutamide) were eligible for the analysis: AFFIRM [43], PREVAIL [44,45], TERRAIN [46], STRIVE [47], OCUU-CRPC [48], PROSPER [49,50], ARCHES [51], ENZAMET [52], CARD [27], NCT02125357 [34]. Investigator-based evaluation of the incidence of cognitive impairment was available in four RCTs (40.0%): one mCRPC, one nmCRPC and two HSPC. All RCTs (100%) included PROs collection, but PRO tools adopted allowed evaluation of cognitive function in one RCT (10.0%). The details of each trial included in the analysis are described below.

In the double-blind, randomized, Phase 3 AFFIRM trial, patients with metastatic CRPC who had progressed after docetaxel treatment were assigned to receive enzalutamide or placebo in addition to ADT. At 12 months, 24.8% of patients treated with enzalutamide were still on treatment. Enzalutamide showed a significant advantage in OS over placebo [43]. In the double-blind, randomized, Phase 3 PREVAIL trial, asymptomatic or mildly symptomatic patients with metastatic chemotherapy-naive CRPC were assigned to receive enzalutamide or placebo [44]. After a mFU of 31 months, 26% of patients in the enzalutamide arm were still on treatment [45]. Enzalutamide showed superior rPFS and OS over placebo. In the AFFIRM and PREVAIL trials, the incidence of investigator-assessed cognitive disorders is not described. PROMs adopted in the studies do not allow evaluation of cognitive functioning [53,54].

In the double-blind, randomized, Phase 2 TERRAIN trial, enzalutamide plus ADT showed superior PFS compared to bicalutamide plus ADT, in asymptomatic or minimally symptomatic patients with metastatic CRPC [46]. After a mFU of 20 months, 32% of patients were still receiving treatment. In the double-blind, randomized, phase II STRIVE trial patients with nonmetastatic or metastatic CRPC were assigned to receive enzalutamide or bicalutamide in addition to ADT [47]. At data cutoff, 53% of patients were still on treatment in the enzalutamide arm. Enzalutamide showed an advantage in the primary endpoint, PFS, versus bicalutamide. In both these studies, the incidence of investigator-assessed cognitive disorders is not available, and PROMs adopted in the studies do not include an evaluation of cognitive function [47,55]. Another randomized Phase 2 trial, OCUU-CRPC, compared enzalutamide to flutamide after combined androgen blockade with bicalutamide [48]. Enzalutamide provided superior clinical outcomes compared to flutamide. Additionally, in this study, the incidence of investigator-assessed cognitive impairment is not available, and PROMs adopted do not allow evaluation of cognitive function.

The double-blind, randomized, Phase 3 PROSPER trial tested enzalutamide versus placebo in patients with nonmetastatic CRPC [49]. After a mFU of 48 months, 41% of patients in experimental the arm were still receiving enzalutamide [50]. Enzalutamide showed a significant advantage in the primary endpoint of metastasis-free survival [49]. Cognitive and memory impairment of any grade (assessed by CTCAE version 4) was detected by investigators in 8% of patients in the enzalutamide arm versus 2% in the placebo arm. The adopted PRO tools did not include an evaluation of cognitive function [56].

The double-blind, randomized, Phase 3 ARCHES trial evaluated enzalutamide versus placebo in addition to ADT in metastatic HSPC [51]. At a mFU of 14.4 months, 76% of patients in the enzalutamide group were still receiving treatment. Enzalutamide showed a significant advantage in the primary endpoint of rPFS. Cognitive/memory impairment of any grade (assessed by CTCAE version 4.03) was detected by investigators in 4.5% in the enzalutamide arm versus 2.1% in the placebo arm, with 0.7% severe events for enzalutamide versus none in the placebo arm. The adopted PRO tools did not include an evaluation of cognitive function [57].

In the open-label, randomized, Phase 3 ENZAMET trial, patients with metastatic HSPC were assigned to receive ADT plus enzalutamide or a standard nonsteroidal antiandrogen therapy [52]. At three years, 62% of patients in the enzalutamide group were still receiving the study drug. Enzalutamide showed an advantage in OS, the primary endpoint. Investigator-assessed cognitive evaluation (graded by CTCAE version 4.02) showed higher rates of any grade cognitive disturbance in the enzalutamide arm (2.8% vs. 0.5%), as well as concentration impairment (5.3% vs. 1.1%), and memory impairment (14.6% vs. 3.6%). No severe cognitive adverse events were detected. Patient-reported cognitive functioning was evaluated with EORTC C30. Kaplan–Meier analysis of deterioration-free survival (DFS) significantly favored enzalutamide vs. placebo (*p* = 0.003, three-year DFS 33% vs. 21%). In our opinion, a methodological issue precludes the correct interpretation of the analysis of time to cognitive function deterioration. The analysis of DFS over three years was performed by the Kaplan–Meier method and log-rank test [58]. However, DFS did not include as event only the worsening of cognitive function, but it was actually a composite endpoint, defined *a priori* as the earliest of death, clinical progression, cessation of study treatment, or a 10-point worsening from baseline (10-point worsening was defined as the minimum clinically important difference). This means that the analysis is largely driven by the differences in outcome, considering that the number of clinical progressions is much higher than the worsening in cognitive function. Consequently, when the authors state that the DFS at three years was better with enzalutamide, this result does not faithfully reflect the real difference in terms of cognitive function. Mean changes in cognitive function were significantly worse with enzalutamide, as documented by the mixed model for repeated measures, and the latter analysis emphasizes the poor interpretability of the former one.

## 10. Trials Testing Both Abiraterone and Enzalutamide

Trials testing both abiraterone and enzalutamide are reported in Table 2.

In the open-label randomized Phase 3 CARD [27], patients with mCRPC who had already received docetaxel and abiraterone or enzalutamide, were randomized to receive cabazitaxel or a second NGHT with the drug not previously received. The primary endpoint was PFS. Patients treated with cabazitaxel showed better PFS and OS compared to patients treated with NGHT. In this study, there is no description of investigator-assessed impairment of cognitive function, and the PROs adopted do not allow evaluation of cognitive function.

The only randomized trial designed to compare abiraterone and enzalutamide was a phase II trial, published in 2019 [34]. The trial enrolled patients eligible for first-line treatment of mCRPC. Description of cognitive function was among the endpoints of the study, along with the evaluation of health-related QoL and depression. Cognitive function was evaluated by investigators by means of MoCA, while the PROs adopted (FACT-P and PHQ9) do not allow evaluation of patient-reported cognitive function. Cognitive screening with MoCA, performed by a trained research nurse at baseline, after 12 weeks of treatment and at treatment discontinuation, did not show significant differences between the two drugs: proportion of patients with a MoCA score <26 (the diagnostic threshold for cognitive impairment) was similar between arms at week 12 (47% abiraterone vs. 54% enzalutamide), 95% CI for difference: −8.6% to 23.8%, *p* = 0.40. In addition, distribution of score change from baseline was similar between arms (*p* = 0.11). However, the sample size was quite small and the trial was probably not powered to detect potentially relevant differences in terms of cognitive function.

Although not randomized, another couple of studies, both conducted in the mCRPC setting, the observational AQUARiUS trial and the phase IV, real-world REAAcT, are worthy of being discussed [39,40,41]. AQUARiUS was designed to describe the impact of abiraterone acetate and enzalutamide on PROs [39,40]. Patients were assessed for a 12-month period from treatment start with the FACT-Cog and EORTC QLQ-C30. The two treatments were compared in terms of “perceived cognitive impairment” and “comments from others” (FACT-Cog) and cognitive functioning from EORTC QLQ-C30, describing the mean difference between groups in the change from baseline and odds ratio of one or more episodes of clinically meaningful worsening. In addition, the two FACT-Cog scales were analyzed describing the time to first PRO showing worsening. Several results related to cognitive impairment were significantly worse in patients receiving enzalutamide. In detail, mean changes in both “perceived cognitive impairment” and “comments from others” (FACT-Cog) and cognitive functioning from EORTC QLQ-C30 were significantly in favor of abiraterone over enzalutamide for three or more consecutive two-month periods up to one year. The number of patients experiencing one or more clinically meaningful worsening episode in cognitive function during the 12 months was significantly lower with abiraterone: 49% vs. 76% in terms of perceived cognitive impairment (odds ratio 0.31, 95% confidence interval 0.14–0.70, *p* = 0.005), 32% vs. 62% in terms of comments from others (odds ratio 0.14, 95%CI 0.05–0.39, *p* < 0.001). For both these FACT-Cog scales, the curves describing the probability of at least one clinically meaningful worsening episode were clearly separated in favor of abiraterone already after two months, remaining separated for the whole period of observation. REAAcT study was designed to evaluate differences in tolerability in patients treated with abiraterone acetate or enzalutamide, limited to the first two months of treatment [41]. Patients were assessed with Cogstate (a standardized, validated computerized test measuring cognitive domains of simple reaction time, choice reaction time, visual episodic memory, and working memory) and several PROs, including FACT-Cog. After two months, Cogstate tests showed no meaningful difference among treatments in the overall mean changes, but a clinically meaningful cognitive decline was observed in 4/46 patients treated with enzalutamide (8.7%) and in 1/46 patients treated with abiraterone (2.2%). Authors report that the overall mean changes from baseline to 2-month assessment in FACT-Cog were similar, without relevant changes, although the proportion of patients showing meaningful improvement in FACT-Cog score was higher with abiraterone (30%) than with enzalutamide (15%), while the proportion of patients with worsening is not reported.

## 11. Trials Testing Apalutamide

Apalutamide is an oral nonsteroidal competitive androgen-receptor inhibitor that binds directly to the ligand-binding domain of the androgen-receptor [59,60].

As shown in Table 4, two RCTs testing apalutamide (2259 patients, 1331 assigned to apalutamide) were eligible for the analysis: SPARTAN [61], TITAN [62]. Investigator-based evaluation of the incidence of cognitive impairment was available in one RCT (50%), conducted in mCRPC. All RCTs (100%) included PROs collection, but PRO tools adopted do not allow evaluation of cognitive function. The details of each trial included in the analysis are described below.

In the double-blind, randomized, Phase 3 SPARTAN trial, patients with nonmetastatic CRPC, with a prostate-specific antigen doubling time of 10 months or less, were assigned to receive apalutamide or placebo, in addition to ADT [61]. After a mFU of 20.3 months, 60.9% of patients were still receiving apalutamide. The addition of apalutamide was associated with a significant improvement in metastasis-free survival.

In the double-blind, randomized, Phase 3 TITAN trial, patients with metastatic hormone-sensitive prostate cancer were assigned to apalutamide or placebo, in addition to ADT [62]. After a mFU of 22.7 months, 66.2% of patients were still receiving apalutamide. The addition of apalutamide was associated with a significant prolongation of PFS and OS.

In the TITAN trial, there are no details about the incidence of investigator-assessed cognitive disorders [62]. In the SPARTAN trial, the incidence of investigator-assessed any grade mental impairment disorder was 5.1% with apalutamide and 3.0% with placebo [61]. Since in both RCTs, QoL was evaluated by the means of FACT-P, no reliable measures of the impact of apalutamide on cognitive function are available [63,64]. Furthermore, no direct comparison is available between apalutamide and the other NGHTs.

## 12. Trials Testing Darolutamide

Darolutamide is an androgen-receptor antagonist with a peculiar structure that is responsible for low binding affinity for γ-aminobutyric acid type A receptors and low penetration of the blood–brain barrier [65].

As shown in Table 5, one RCT testing darolutamide (1509 patients, 955 assigned to apalutamide) was eligible for the analysis: ARAMIS [66].

In the double-blind, randomized, Phase 3 ARAMIS trial, patients with nonmetastatic CRPC, with a prostate-specific antigen doubling time of 10 months or less, were randomly assigned to receive darolutamide or placebo, in addition to ADT [66]. After a mFU of 17.9 months, 64% of patients were still receiving darolutamide. The addition of darolutamide was associated with a significant improvement in metastasis-free survival.

In the description of investigator-assessed adverse events, no significant difference in the incidence of any grade cognitive disorders was described between darolutamide and placebo (0.4% vs. 0.2% respectively). However, given that QoL was evaluated by the means of FACT-P and EORTC PR25, no reliable measure of the impact of darolutamide on cognitive function is available [66]. Furthermore, no direct comparison is available between darolutamide and the other NGHTs.

## 13. Conclusions

When planning this systematic review, our aim was to describe the evidence about cognitive function in all the clinical settings, although heterogeneous (metastatic castration-resistant, nonmetastatic castration-resistant, hormone-sensitive), where new-generation hormonal treatments have produced positive results, receiving approval for use in clinical practice. Despite patients’ QoL and the balance between disease control and treatment adverse events are widely recognized as relevant clinical issues in patients with prostate cancer, this systematic review shows that the clinical development of NGHT, in all the clinical settings, has not included a complete, systematic evaluation of cognitive function.

Commonly used QoL instruments do not allow (FACT-P) or allow only partially (EORTC QLQ C30) a description of the trajectory of cognitive function. On the other hand, FACT-Cog has been specifically developed to assess cognitive function and, if used in addition to other commonly adopted QoL instruments, its use could allow a homogeneous evaluation in clinical trials. This should be adequately considered when planning trial design, endpoints and instruments in this setting.

The few available data suggest that cognitive deterioration could be clinically relevant, at least in a proportion of patients. Indirect comparison among clinical trials suggests that there could be differences between drugs. For instance, darolutamide has a lower blood–brain barrier penetration compared to other drugs, and this, at least in principle, could imply a lower incidence of CNS adverse events. However, the absence of direct comparisons does not help to obtain solid evidence.

Attention to cognitive deterioration should be included in the clinical management of these patients, especially considering the long duration of treatment expected in “earlier” settings, like the treatment of hormone-sensitive disease.

The attention to QoL of patients with prostate cancer needs to be improved in the near future, in order to become a real *fil rouge* for clinical research and clinical practice. In future trials, cognitive status should be evaluated with validated tools.

## Figures and Tables

**Figure 1 cancers-12-02568-f001:**
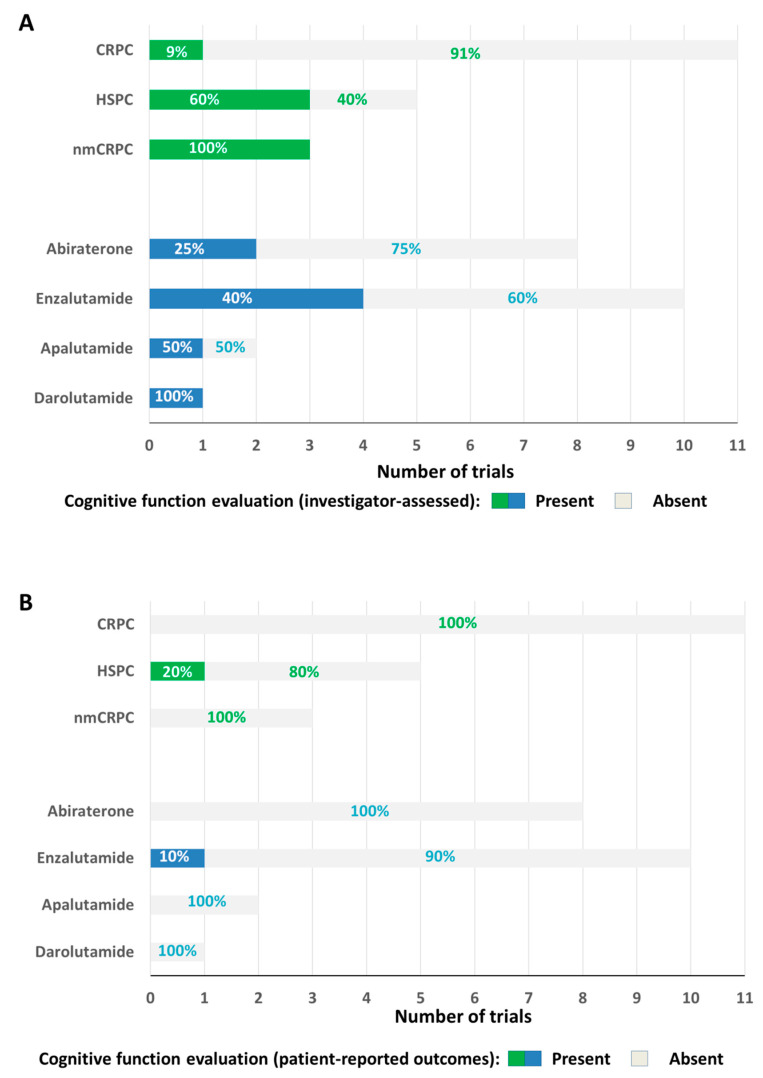
Availability of evaluation of cognitive function based on investigator assessment (**A**) and patient-reported outcomes (**B**) in the randomized trials (RCTs) of new-generation hormonal therapies.

**Table 1 cancers-12-02568-t001:** Trials testing abiraterone.

Study	Setting	Comparison	Investigator-Assessed Evaluation of Cognitive Function	PRO Tools Adopted in the Study	PRO-Based Evaluation of Cognitive Function
COU-AA-301 [8,28,35]	Postdocetaxel CRPC	Randomized trial (abiraterone + prednisone + ADT vs. placebo + prednisone + ADT)	Not available	FACT-P BPI-SF BFI	Not available (the adopted PRO tools do not include an evaluation of cognitive function)
COU-AA-302 [9,29,36]	Chemotherapy-naïve CRPC	Randomized trial (abiraterone + prednisone + ADT vs. placebo + prednisone + ADT)	Not available	FACT-P BPI-SF	Not available (the adopted PRO tools do not include an evaluation of cognitive function)
NCT01695135 [30]	Postdocetaxel CRPC	Randomized trial (abiraterone + prednisone + ADT vs. placebo + prednisone + ADT)	Not available	FACT-P BPI-SF BFI-SF	Not available (the adopted PRO tools do not include an evaluation of cognitive function)
NCT01591122 [31]	CRPC	Randomized trial (abiraterone + prednisone + ADT vs. placebo + prednisone + ADT)	Not available	FACT-P BPI-SF	Not available (the adopted PRO tools do not include an evaluation of cognitive function)
STAMPEDE [32,37]	HSPC	Randomized trial (abiraterone + prednisolone + ADT vs. ADT alone)	Cognitive disturbance: Any grade: abiraterone 6.4%; control 3.8% Grade 3–4: abiraterone 0.4%; control 0.2%	EORTC C30 EORTC P25	Not available (EORTC C30 includes evaluation of cognitive functioning but results are not reported in the publication)
LATITUDE [10,33,38]	HSPC	Randomized trial (abiraterone + prednisone + ADT vs. placebo + prednisone +ADT)	Not available	FACT-P BPI-SF BFI EQ-5D-5L	Not available (the adopted PRO tools do not include an evaluation of cognitive function)

PRO: patient-reported outcome; CRPC: Castration-resistant prostate cancer; HSPC: hormone-sensitive prostate cancer; ADT: androgen deprivation therapy; FACT-P: Functional Assessment of Cancer Therapy-Prostate; BPI-SF: Brief Pain Inventory-Short Form; BFI: Brief Fatigue Inventory; EORTC QLQ-C30: European Organization for Research and Treatment of Cancer-Quality of Life Questionnaire-Core 30; EORTC QLQ-PR25; European Organization for Research and Treatment of Cancer-Quality of Life Questionnaire-Prostate Cancer 25; EQ-5D: EuroQoL-5 dimensions.

**Table 2 cancers-12-02568-t002:** Trials testing both abiraterone and enzalutamide.

Study	Setting	Comparison	Investigator-Assessed Evaluation of Cognitive Function	PRO Tools Adopted in the Study	PRO-Based Evaluation of Cognitive Function
CARD [27]	CRPC	Randomized trial (Cabazitaxel vs. abiraterone + prednisone or enzalutamide; drug in the arm randomized to hormonal treatment was at Investigator’s discretion)	Not available	FACT-P BPI-SF EQ-5D-5L	Not available (the adopted PRO tools do not include evaluation of cognitive function)
NCT02125357 [34]	CRPC	Randomized trial (Abiraterone + prednisone vs. Enzalutamide)	Montreal Cognitive Assessment (screening test): Proportion of patients with a MoCA score < 26 similar between arms at week 12 (47% abiraterone vs. 54% enzalutamide), 95% CI for difference: −8.6% to 23.8%, *p* = 0.4; Distribution of score change from baseline was similar between arms (*p* = 0.11).	FACT-P PHQ-9	Not available (the adopted PRO tools do not include evaluation of cognitive function)
AQUARiUS [39,40]	CRPC	Abiraterone + prednisone vs. Enzalutamide (observational, non randomized study)	Not available	FACT-Cog EORTC QLQ- C30 BFI-SF BPI-SF	Scales: Cognitive functioning (EORTC QLQ C30) Perceived cognitive impairment (FACT-Cog) Comments from others (FACT-Cog) Mean difference between groups in the change from baseline: For all 3 scales, significantly in favour of abiraterone over enzalutamide for 3 or more consecutive periods in the 12 months Proportion of patients with one or more episodes of clinically meaningful worsening in each group: Perceived cognitive impairment: 49% abiraterone vs. 76% enzalutamide (odds ratio 0.31, 95% confidence interval 0.14–0.70, *p* = 0.005) Comments from others: 32% abiraterone vs. 62% enzalutamide (odds ratio 0.14, 95%CI 0.05–0.39, *p* < 0.001). Cognitive functioning: statistically significant in favour of abiraterone, numbers not reported. Time to worsening: Kaplan-Meier curves describing the probability of at least 1 clinically meaningful worsening episode were clearly separated in favour of abiraterone for both FACT-Cog items
REAAcT [41]	CRPC	Abiraterone + prednisone vs. Enzalutamide (observational, non randomized study)	Cogstate tests: mean changes from baseline were similar for both arms and showed no meaningful change over the first 2 months of treatment; clinically meaningful cognitive decline in 1/46 pts (2.2%) with abiraterone and 4/46 pts (8.7%) with enzalutamide.	FACT-Cog FACIT-Fatigue EORTC QLQ-C30	Mean change in FACT-Cog total score: 0.22 with abiraterone vs. −3.34 with enzalutamide (p value: not available); Proportions of patients with improvement (minimum clinically important difference) on FACT-Cog score: 30% with abiraterone vs. 15% with enzalutamide (p value: not available).

PRO: patient-reported outcome; CRPC: Castration-resistant prostate cancer; MoCA: Montreal Cognitive Assessment; FACT-P: Functional Assessment of Cancer Therapy-Prostate; FACT-Cog: Functional Assessment of Cancer Therapy-Cognitive; BPI-SF: Brief Pain Inventory-Short Form; BFI-SF: Brief Fatigue Inventory-Short Form; EORTC QLQ-C30: European Organization for Research and Treatment of Cancer-Quality of Life Questionnaire-Core 30; EQ-5D: EuroQoL-5 dimensions; PHQ-9: Patient Health Questionnaire-9; FACIT-Fatigue: Functional Assessment of Chronic Illness Therapy-Fatigue.

**Table 3 cancers-12-02568-t003:** Trials testing enzalutamide.

Study	Setting	Comparison	Investigator-Assessed Evaluation of Cognitive Function	PRO Tools Adopted in the Study	PRO-Based Evaluation of Cognitive Function
AFFIRM [43,53]	Postchemotherapy CRPC	Randomized trial (enzalutamide + ADT vs. placebo + ADT)	Not available	FACT-P BPI-SF EQ5D	Not available (the adopted PRO tools do not include an evaluation of cognitive function)
PREVAIL [44,45,54]	Chemotherapy-naïve CRPC	Randomized trial (enzalutamide + ADT vs. placebo + ADT)	Not available	FACT-P BPI-SF EQ5D	Not available (the adopted PRO tools do not include an evaluation of cognitive function)
TERRAIN [46,55]	Chemotherapy-naïve CRPC	Randomized trial (enzalutamide + placebo + ADT vs. placebo + bicalutamide + ADT)	Not available	FACT-P BPI-SF EQ5D	Not available (the adopted PRO tools do not include an evaluation of cognitive function)
STRIVE [47]	Chemotherapy-naïve CRPC	Randomized trial (enzalutamide + ADT vs. bicalutamide + ADT)	Not available	FACT-P	Not available (the adopted PRO tools do not include an evaluation of cognitive function)
OCUU-CRPC [48]	CRPC after ADT + bicalutamide	Randomized trial (enzalutamide + ADT vs. flutamide + ADT)	Not available	FACT-P	Not available (the adopted PRO tools do not include an evaluation of cognitive function)
PROSPER [49,50,56]	Nonmetastatic CRPC	Randomized trial (enzalutamide + ADT vs. placebo + ADT)	Cognitive and memory impairment (disturbance in attention, cognitive disorders, amnesia, Alzheimer’s disease, dementia, senile dementia, mental impairment and vascular dementia): Any grade: enzalutamide 8%; placebo 2%.	FACT-P EORTC QLQ PR25 BPI-SF EQ5D	Not available (the adopted PRO tools do not include an evaluation of cognitive function)
ARCHES [51,57]	HSPC	Randomized trial (enzalutamide + ADT vs. placebo + ADT)	Cognitive/memory impairment: Any grade: enzalutamide 4.5%; placebo 2.1%. Grade 3–4: enzalutamide 0.7%; placebo 0%.	BPI-SF FACT-P EORTC QLQ PR25 EQ5D	Not available (the adopted PRO tools do not include an evaluation of cognitive function)
ENZAMET [52,58]	HSPC	Randomized trial (enzalutamide + ADT +/– docetaxel vs. standard nonsteroidal antiandrogen therapy + ADT +/– docetaxel	Cognitive disturbance: Any grade: enzalutamide 2.8%; SoC 0.5%. Grade 3–4: enzalutamide 0%; SoC 0%. Concentration impairment: Any grade: enzalutamide 5.3%; SoC 1.1%. Grade 3–4: enzalutamide 0%; placebo 0%. Memory impairment: Any grade: enzalutamide 14.6%; SoC 3.6%. Grade 3–4: enzalutamide 0%; SoC 0%.	EORTC QLQ C30 EORTC PR25 EQ5D	Scale: Cognitive functioning (EORTC QLQ C30) Deterioration-free survival at 3 years: Kaplan–Meier analysis favors enzalutamide vs. placebo (33% vs. 21%, *p* = 0.0003) Mean changes in cognitive function: significantly worse with enzalutamide (least squares mean difference 3.9, 95%CI 2.4–5.4, *p* < 0.0001)

PRO: patient-reported outcome; CRPC: Castration-resistant prostate cancer; HSPC: hormone-sensitive prostate cancer; ADT: androgen deprivation therapy; SoC: standard of care; FACT-P: Functional Assessment of Cancer Therapy-Prostate; BPI-SF: Brief Pain Inventory-Short Form; BFI: Brief Fatigue Inventory; EORTC QLQ-C30: European Organization for Research and Treatment of Cancer-Quality of Life Questionnaire-Core 30; EORTC QLQ-PR25; European Organization for Research and Treatment of Cancer-Quality of Life Questionnaire-Prostate Cancer 25; EQ-5D: EuroQoL-5 dimensions.

**Table 4 cancers-12-02568-t004:** Trials testing apalutamide.

Study	Setting	Comparison	Investigator-Assessed Evaluation of Cognitive Function	PRO Tools Adopted in the Study	PRO-Based Evaluation of Cognitive Function
SPARTAN [61,63]	Nonmetastatic CRPC	Randomized trial (apalutamide + ADT vs. placebo + ADT)	Mental impairment disorders (disturbance in attention, memory impairment, cognitive disorder and amnesia): Any grade: apalutamide 5.1%; placebo 3.0%. Grade 3–4: apalutamide 0; placebo 0.	FACT-P EQ5D	Not available (the adopted PRO tools do not include an evaluation of cognitive function)
TITAN [62,64]	HSPC	Randomized trial (apalutamide + ADT vs. placebo + ADT)	Not available	BPI-SF BFI FACT-P EQ5D	Not available (the adopted PRO tools do not include an evaluation of cognitive function)

PRO: patient-reported outcome; CRPC: Castration-resistant prostate cancer; HSPC: hormone-sensitive prostate cancer; ADT: androgen deprivation therapy; FACT-P: Functional Assessment of Cancer Therapy-Prostate; BPI-SF: Brief Pain Inventory-Short Form; BFI: Brief Fatigue Inventory; EQ-5D: EuroQoL-5 dimensions.

**Table 5 cancers-12-02568-t005:** Trials testing darolutamide.

Study	Setting	Comparison	Investigator-Assessed Evaluation of Cognitive Function	PRO Tools Adopted in the Study	PRO-Based Evaluation of Cognitive Function
ARAMIS [66]	Nonmetastatic CRPC	Randomized trial (darolutamide + ADT vs. placebo + ADT)	Any grade cognitive disorders: darolutamide 0.4%; placebo 0.2% Grade 3–4 cognitive disorders: darolutamide 0; placebo 0 Any grade memory impairment: darolutamide 0.5%; placebo 1.3% Grade 3–4 memory impairment: darolutamide 0; placebo 0	FACT-P EORTC QLQ PR25 EQ5D	Not available (FACT-P and EORTC PR-25 do not include an evaluation of cognitive function)

PRO: patient-reported outcome; CRPC: Castration-resistant prostate cancer; ADT: androgen deprivation therapy; FACT-P: Functional Assessment of Cancer Therapy-Prostate; EORTC QLQ-PR25; European Organization for Research and Treatment of Cancer-Quality of Life Questionnaire-Prostate Cancer 25; EQ-5D: EuroQoL-5 dimensions.

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
