# Peer review of "Evaluation of Cognitive Function in Trials Testing New-Generation Hormonal Therapy in Patients with Prostate Cancer: A Systematic Review"

_cancers, 2020, doi:10.3390/cancers12092568_

Round 1

Reviewer 1 Report

The article makes a comprehensive review of the studies published to date with the new hormonal agents in different scenarios in metastatic prostate cancer. The current review digresses more into the data of the studies (endpoints, median duration of treatment, etc) than the point this review actually aims to analyze, which is cognitive impairment resulting from the treatments. The truth is that the existing data on this subject are very limited and probably not enough importance has been given to this side effect in the different studies. 
The methods section would not leave it to the end, it would place between line 176 and 177 as it is essential to know what the search criteria and exclusion criteria have been used in the present analysis.
In lines 275 - 277 there are differences in the size of the characters and I think also in the font of the text.

Author Response

Reviewer 1

The article makes a comprehensive review of the studies published to date with the new hormonal agents in different scenarios in metastatic prostate cancer. The current review digresses more into the data of the studies (endpoints, median duration of treatment, etc) than the point this review actually aims to analyze, which is cognitive impairment resulting from the treatments. The truth is that the existing data on this subject are very limited and probably not enough importance has been given to this side effect in the different studies. 

Re: we agree with the Reviewer, the main point of our review is that existing data on this topic are disappointingly limited, as we also emphasized in the Conclusions. We decided to give some details about the studies (particularly the median duration of follow-up and the number of subjects still on treatment at the time of analysis, for 2 reasons:

  • To underline that the incidence of cognitive impairment is probably underestimated also due to the limited study follow-up
  • To emphasize the increasing duration of treatment with hormonal agents, moving from castration-resistant setting to non-metastatic patients and to hormone-sensitive patients. This could reasonably increase the cumulative risks of treatment-emergent cognitive impairment.  

The Methods section would not leave it to the end, it would place between line 176 and 177 as it is essential to know what the search criteria and exclusion criteria have been used in the present analysis. 

Re: we thank the Reviewer for this comment. We had placed the Methods (including the Literature search) at the end of the text according to journal’s instructions, however, if the Editor agrees, we have moved that paragraph in the suggested position, before the description of the search results.

In lines 275 - 277 there are differences in the size of the characters and I think also in the font of the text.

Re: we modified the size and font of characters, now it is uniform.

Reviewer 2 Report

The manuscript by Marandino, et al. is a systemic review aiming to evaluate the impact of hormone therapy on verbal memory and executive function in men with prostate cancer. 19 randomized control trial studies were considered in this review. Understanding and analyzing the incidence, severity of cognitive impairments and their impact on quality of life in patients with advance disease like prostate cancer is challenging; however, this comprehensive review provides a glimpse of  information which might help the physicians in the choice of treatment. The only comment and/or suggestion that will help the readers in this area is that the authors are requested to provide some graphical representation of the summary of their results or the trials included in this manuscript. Like with a pie-chart, etc.

Reviewer 3 Report

The authors performed a systematic review of literature, reviewing the role of new-generation hormonal therapy (NGHT), added to androgen deprivation therapy on cognitive function.  it is an interesting and often underestimated topic both by research and by the potential implications for clinical practice. The Authors concluded that the evaluation of  cognitive function should be included in clinical management of these patients, since the cognitive deterioration could be clinically relevant

Comments

-the search strategy has been inserted at the end of the text. to better understand your work, the search strategy should be inserted at the beginning of the text immediately after the introductory part

-Previously research in the field have clearly showed conflictual results: several prospective and retrospective trials does not showed that ADT use was associated to cognitive decline. Moreover two systematic review and metanalysis fail to demonstrate a role of ADT in cognitive function impairment. I believe that this aspect should be more stressed in the introduction section. Based on previously research article in this field, your research question could be…”Which is the add value of  NHGT on cognitive function?

-your analysis clearly shows that the assessment of cognitive function is underestimated in most clinical trials. Although few trials have evaluated cognitive function, there seems to be a difference in cognitive impairment between different new generation drugs with daralutamide which seems to have a lower impact than enzalutamide and abiraterone. how to interpret this data? can it be linked to the mechanism of action of the drug?

I think this should be reported in the discussion to better guide future research in this field

-In your analysis, data on cognitive function for abiraterone and enzalutamide are only available for a specific setting of patients (HSPC an CRPC). Based on your data, we don’t know if the stage of disease could play a role in cognitive decline: for example, a patient with mCRCP is more at risk of having a cognitive decline than a patient with HSPC? I think this aspect is important for clinical practice and also for further research.

Moreover for other type of cancer like breast , we know that chemotherapy can impact negatively on cognitive function through a neurotoxic effect. this aspect has never been evaluated in patients with prostate cancer treated with docetaxel or cabazitaxel. and I believe it can influence the interpretation of the data on the role of NHGT

-your analysis showed substantial heterogeneity both in the patient populations analyzed and in the tools used to assess cognitive function. This aspect should be reported in your conclusions. Based in your experience in the field, which tool should be used in future clinical trials? 

-Given the conflicting results on the role of ADT in cognitive decline and the data that PRO-based cognitive function results were presented only in 1 RCT (5.3%), I believe that a reasonable conclusion of your work could be….. "the evaluation of cognitive function with validated tools is lacking in the trials available. Furthermore, the fact that some studies indicate a possible role of NHGT in cognitive decline, cognitive status should be evaluated with validated tools in future trials

Author Response

Point-by-point response to Reviewer 3 comments.

Reviewer 3

The authors performed a systematic review of literature, reviewing the role of new-generation hormonal therapy (NGHT), added to androgen deprivation therapy on cognitive function.  it is an interesting and often underestimated topic both by research and by the potential implications for clinical practice. The Authors concluded that the evaluation of  cognitive function should be included in clinical management of these patients, since the cognitive deterioration could be clinically relevant

Comments

 -the search strategy has been inserted at the end of the text. to better understand your work, the search strategy should be inserted at the beginning of the text immediately after the introductory part

Re: thanks for the comment. As explained in the reply to Reviewer 1, Methods had been placed at the end of the text following journal’s instructions. We have moved before the description of results.

-Previously research in the field have clearly showed conflictual results: several prospective and retrospective trials does not showed that ADT use was associated to cognitive decline. Moreover two systematic review and metanalysis fail to demonstrate a role of ADT in cognitive function impairment. I believe that this aspect should be more stressed in the introduction section. Based on previously research article in this field, your research question could be…”Which is the add value of  NHGT on cognitive function?

Re: we thank the Reviewer for this comment. We already had cited in the Introduction a couple of articles discussing the conflicting evidence about ADT and cognitive decline (reference 3, reference 4), explicitly stating that the evidence about cognitive impairment in patients receiving ADT is variable:

Although the evidence about cognitive impairment in patients receiving ADT is variable, several studies have shown that ADT can produce a negative effect on cognitive functioning [4-6].

Following Reviewer’s comment, we added another sentence to reinforce this concept:

Overall, results of studies trying to describe the association between ADT and both cognitive changes and other CNS effects in patients with prostate are controversial [3].

-your analysis clearly shows that the assessment of cognitive function is underestimated in most clinical trials. Although few trials have evaluated cognitive function, there seems to be a difference in cognitive impairment between different new generation drugs with daralutamide which seems to have a lower impact than enzalutamide and abiraterone. how to interpret this data? can it be linked to the mechanism of action of the drug? I think this should be reported in the discussion to better guide future research in this field

Re: this is an interesting comment. Darolutamide has negligible blood-brain barrier penetration as demonstrated in mouse pharmacokinetics studies, with a brain/plasma ratio of about 2% compared to 25% for enzalutamide. At least in theory, this may imply improved central nervous system (CNS) potential adverse events. Of course the absence of direct comparisons does not help to obtain solid evidence. However, we added a sentence in the Conclusions about this topic:

Indirect comparison among clinical trials suggests that there could be differences between drugs. For instance, darolutamide has a lower blood-brain barrier penetrationcompared to other drugs, and this, at least in principle, could imply lower incidence of CNS adverse events.   However, the absence of direct comparisons does not help to obtain solid evidence.

-In your analysis, data on cognitive function for abiraterone and enzalutamide are only available for a specific setting of patients (HSPC an CRPC). Based on your data, we don’t know if the stage of disease could play a role in cognitive decline: for example, a patient with mCRCP is more at risk of having a cognitive decline than a patient with HSPC? I think this aspect is important for clinical practice and also for further research.

Re: this is an interesting comment. Patients in the different settings (mCRPC, nmCRPC, HSPC) on average are different for type of previous treatments, duration of previous androgen deprivation therapy, expected duration of new-generation hormonal treatments. This could have a significant impact on the incidence and severity of cognitive impairment. The best way to answer to this interesting question would be the availability of robust data from randomized clinical trials that, unfortunately, are not available.

Moreover for other type of cancer like breast , we know that chemotherapy can impact negatively on cognitive function through a neurotoxic effect. this aspect has never been evaluated in patients with prostate cancer treated with docetaxel or cabazitaxel. and I believe it can influence the interpretation of the data on the role of NHGT

Re: this is an interesting comment. Our aim was to focus on hormonal treatment, so our literature search and our synthesis of results is specifically limited to hormonal drugs. However, we agree that chemotherapy (received by patients in the castration-resistant setting and in some cases also in the hormone-sensitive disease) could have an impact on patients’ cognitive deterioration. This could happen both before and after the use of new generation hormonal drugs, based on the treatment sequence used in each patient. If the first trials with new generation hormonal drugs were conducted in chemotherapy-pretreated patients, the majority of patients in the most recent trials were chemotherapy-naïve. We added a sentence in the Introduction about this issue:

In addition to ADT, also taxane-based chemotherapy, that is commonly used in patients with metastatic castration-resistant prostate cancer (CRPC) and – more recently – also in the hormone-sensitive setting, could have a negative impact on cognitive function (https://pubmed.ncbi.nlm.nih.gov/32352155/).

-your analysis showed substantial heterogeneity both in the patient populations analyzed and in the tools used to assess cognitive function. This aspect should be reported in your conclusions. Based in your experience in the field, which tool should be used in future clinical trials? 

Re: we thank the Reviewer for this comment. As for patient populations, when planning this systematic review we decided to describe results obtained in all the clinical settings – although heterogeneous - where new generation hormonal treatments have produced positive results receiving approval for use in clinical practice. As for the assessment of cognitive function, the problem is that most PRO tools used in clinical practice are NOT adequate to assess cognitive function. As described in the dedicated paragraph, FACT-Cog is specifically developed to assess cognitive function and, used in addition to common PRO instruments like EORTC or FACT G – FACT P, its use could allow a homogeneous evaluation in clinical trials. As suggested by the Reviewer, we added this concepts in the Conclusions:

“When planning this systematic review, our aim was to describe the evidence about cognitive function in all the clinical settings – although heterogeneous – (metastatic castration-resistant, non metastatic castration resistant, hormone sensitive) where new generation hormonal treatments have produced positive results, receiving approval for use in clinical practice. Despite patients’ QoL and the balance between disease control and treatment adverse events are widely recognized as relevant clinical issues in patients with prostate cancer, this systematic review shows that the clinical development of NGHT, in all the clinical settings, has not included a complete, systematic evaluation of cognitive function.

Commonly used QoL instruments do not allow (FACT-P) or allow only partially (EORTC QLQ C30) a description of the trajectory of cognitive function. On the other hand, FACT-Cog has been specifically developed to assess cognitive function and, if used in addition to other commonly adopted QoL instruments, its use could allow a homogeneous evaluation in clinical trials. This should be adequately considered when planning trial design, endpoints and instruments in this setting.” 

-Given the conflicting results on the role of ADT in cognitive decline and the data that PRO-based cognitive function results were presented only in 1 RCT (5.3%), I believe that a reasonable conclusion of your work could be….. "the evaluation of cognitive function with validated tools is lacking in the trials available. Furthermore, the fact that some studies indicate a possible role of NHGT in cognitive decline, cognitive status should be evaluated with validated tools in future trials

Re: we thank the Reviewer for this comment. We modified Conclusions to incorporate all the Reviewers’ comments, and these concepts are explicitly included. Last sentence of the Conclusions is :

In future trials, cognitive status should be evaluated with validated tools.   

Round 2

Reviewer 3 Report

The manuscript was extensively revised in accordance with the reviewer's requests. 

I believe that the state of the manuscript has improved and I have no other requests to make